# Structural Contour Map of the Iota Carbonic Anhydrase from the Diatom *Thalassiosira pseudonana* Using a Multiprong Approach

**DOI:** 10.3390/ijms22168723

**Published:** 2021-08-13

**Authors:** Erik L. Jensen, Véronique Receveur-Brechot, Mohand Hachemane, Laura Wils, Pascale Barbier, Goetz Parsiegla, Brigitte Gontero, Hélène Launay

**Affiliations:** 1Aix Marseille Univ, CNRS, BIP, UMR 7281, IMM, FR 3479, 31 Chemin J. Aiguier, CEDEX 20, 13 402 Marseille, France; jensen@ibpc.fr (E.L.J.); veronique.brechot@imm.cnrs.fr (V.R.-B.); mohand-said.HACHEMANE@etu.univ-amu.fr (M.H.); laura.wils@hotmail.fr (L.W.); goetz.parsiegla@imm.cnrs.fr (G.P.); 2Aix Marseille Univ, CNRS, INP, Inst Neurophysiopathol, 13 402 Marseille, France; pascale.barbier@univ-amu.fr

**Keywords:** analytical ultracentrifugation, CO_2_ concentrating mechanism, diffusion-ordered NMR spectroscopy, electrospray ionization mass spectrometry, homotetramer, manganese, metalloprotein, photosynthesis, small-angle X-ray scattering

## Abstract

Carbonic anhydrases (CAs) are a family of ubiquitous enzymes that catalyze the interconversion of CO_2_ and HCO_3_^−^. The “iota” class (ι-CA) was first found in the marine diatom *Thalassiosira pseudonana* (tpι-CA) and is widespread among photosynthetic microalgae and prokaryotes. The ι-CA has a domain COG4875 (or COG4337) that can be repeated from one to several times and resembles a calcium–calmodulin protein kinase II association domain (CaMKII-AD). The crystal structure of this domain in the ι-CA from a cyanobacterium and a chlorarachniophyte has been recently determined. However, the three-dimensional organization of the four domain-containing tpι-CA is unknown. Using biophysical techniques and 3-D modeling, we show that the homotetrameric tpι-CA in solution has a flat “drone-like” shape with a core formed by the association of the first two domains of each monomer, and four protruding arms formed by domains 3 and 4. We also observe that the short linker between domains 3 and 4 in each monomer confers high flexibility, allowing for different conformations to be adopted. We propose the possible 3-D structure of a truncated tpι-CA containing fewer domain repeats using experimental data and discuss the implications of this atypical shape on the activity and metal coordination of the ι-CA.

## 1. Introduction

Carbonic anhydrases (CAs; EC 4.2.1.1) are widespread enzymes found in all domains of life [1,2]. They all catalyze the same reversible reaction of CO_2_ hydration to form HCO_3_^−^. Several classes of CAs have been described so far, named using the Greek letters α-, β-, γ-, δ-, ζ-, η-, θ- and ι- [1,3,4,5,6]. CAs participate in numerous cellular processes, such as pH regulation, ion transport and cell metabolism [7], and in CO_2_ concentrating mechanisms in photosynthetic organisms [2,8]. Living organisms may possess one to several different CA classes encoded in their genomes, reflecting the diversity of CAs [2,8,9].

Until recently, nearly all CAs have been described as metalloenzymes, which commonly use Zn^2+^ as a metal cofactor; however, some CAs from the γ-, δ- and ζ- classes are cambialistic and are able to replace Zn^2+^ by Fe^2+^, Co^2+^ and Cd^2+^, respectively [4,10]. Although the different CA classes catalyze the same reaction, they share little or no apparent evolutionary relationship in terms of amino acid sequence or structure [1], including the amino acids involved in the coordination of their metal ion cofactor and catalytic site as well as their oligomeric state [7,11].

The most recently described CAs from the new ι- class was first found in the marine diatom *Thalassiosira pseudonana* [6,12]. This ι-CA is an important component of the diatom CO_2_-concentrating mechanisms (CCMs) that are essential for carbon fixation in many photosynthetic organisms [13]. In addition, ι-CAs are widespread among marine photosynthetic microalgae and non-photosynthetic prokaryotes, which suggest an important role of this CA in global carbon biogeochemical cycling [6]. The ι-CAs have metal cofactors that differ from one organism to the other. While ι-CA from *T. pseudonana* was shown to use Mn^2+^ instead of Zn^2+^, a recently reported ι-CA homolog from the bacterium *Burkholderia territorii* is highly specific for Zn^+2^ [14]. Moreover, a novel type of ι-CA from a cyanobacterium (*Anabaena* sp. PCC7120) and a chlorarachniophyte alga (*Bigelowiella natans*) that is able to act without any metal cofactor was recently described [15]. These differences in metal cofactor preference of ι-CA homologs might be related to specific features of their structure. The amino acid sequence of the ι-CA from diatoms is characterized by a domain (COG4875) from the nuclear transport factor 2 (NTF2) family, which is found in proteins with a broad range of biological functions [16]. This domain is also highly similar to the calcium-calmodulin protein kinase II association domain (CaMKII-AD) and can be repeated one to several times along the protein sequence. The amino acid sequence of the ι-CA from *T. pseudonana* has four domain repetitions, but in other algal species, it can contain two or three (or more) repetitions [6,15]. Interestingly, most sequences from prokaryotes contain only one domain—in contrast to sequences from eukaryotes—which may reflect an evolutionary trait in species from different domains in the Tree of Life. The CaMKII-AD has a known role in protein oligomerization [17] as well as in NTF2 proteins, which are known to form dimers [18], and in both cases, the domain structure is characterized by a cone-shaped cavity formed by an angled arrangement of a β-sheet and α-helices [16,19]. A similar structure has been predicted from the dimeric ι-CA from *B. territorii*, which contains one domain [14]. Similarly, the X-ray crystal structure of the COG4337 domains that compose the ι-CA from *Anabaena* and *B. natans* confirmed a high structural homology with the CaMKII-AD from *Xanthomonas campestris* (PDB: 3H51) [15]. However, there is still not a resolved structure from a full-length multi-domain-containing ι-CA from diatoms based on experimental data. This information could help to determine the 3D arrangement of multiple CaMKII-AD-containing proteins as well as their multimeric organization.

In this work, we describe some structural features of the four-domain-repeats containing ι-CA from *T. pseudonana* (tpι-CA) using different biophysical techniques. We constructed a model of the homotetrameric form of tpι-CA using predicted 3D models integrating small-angle X-ray scattering (SAXS) and nuclear magnetic resonance (NMR) approaches. In addition, based on these data, we also proposed a model for other ι-CAs that contain three or fewer domain repetitions.

## 2. Results

### 2.1. Oligomerization State of the tpι-CA

Two oligomeric forms of the recombinant tpι-CA in solution have previously been observed [6] (Figure 1b), which were named the “high molecular mass” (HMM) and “low molecular mass” (LMM) forms as their real molecular masses were not determined. The elution volumes of both forms on size exclusion chromatography (SEC) were much smaller than expected for tpι-CA monomers, indicating a higher oligomerization state for both forms. Congo red spectral shift assay experiments were used to exclude the possibility that the HMM form resulted from a denatured and amyloid-like aggregated form of tpι-CA. Our results showed that Congo red does not bind to tpι-CA and, thus, suggest that the protein does not form fibrils in solution in our conditions (Figure 1a). Besides an absorption at 280 nm, the HMM form unexpectedly also showed an absorption at 260 nm, indicating the presence of nucleic acid in this sample. Agarose gel electrophoresis and ethidium bromide staining confirmed the presence of DNA or RNA in this form (not shown). Using SEC, we showed that, when the protein sample was treated with Benzonase, a nuclease that attacks and degrades all forms of nucleic acids, the amount of HMM form decreased while the amount of LMM increased (Figure 1b). We speculate that this DNA/RNA binding is very likely to be unspecific and not physiological relevant because tpι-CA is located in the vicinity of chloroplast membranes [6]. Consequently, further structural characterization was performed instead on the LMM form using dynamic light scattering (DLS) and analytical ultracentrifugation (AUC). We observed that this LMM form was principally monodisperse. The LMM form has an hydrodynamic radius of 8.81 ± 0.4 nm, determined by DLS, and a sedimentation coefficient standard S^0^_20,W_ (20 °C in water and extrapolated to protein concentration equal to zero) of 8.5 S, determined by AUC (Figure 1c,d).

Electrospray ionization mass spectrometry (ESI-MS) was used under non-denaturing conditions to probe the oligomerization state of the tpι-CA in solution. We observed a distribution of multiple charged ions of tpι-CA from 28 to 32 that corresponded to a calculated averaged neutral molecular mass of 260 kDa, indicating that the oligomerization state of tpι-CA is a homotetramer (Table 1). In addition, after glutaraldehyde-induced protein cross-linking, SDS-PAGE and Western blot analysis also showed the presence of a homotetrameric form of ~240 kDa, together with possible trimeric and dimeric intermediates (~120 and ~180 kDa, respectively; Figure 2). The homotetrameric state of tpι-CA is also in agreement with the observed apparent molecular mass of 280 kDa (Figure 1b) observed by SEC.

### 2.2. Characterization of the Secondary Structure of tpι-CA and Its Domain Variants

As previously described, the ι-CA is widely distributed among living organisms, and the number of COG4875 domain repetitions contained within its sequence may vary among different species [6]. The ι-CA protein from *T. pseudonana* contains four repetitions of the COG4875 domain along its full amino acid sequence, excluding a chloroplast-targeting signal peptide on its N-terminus. These four COG4875 domains shared more than 60% of amino acid identity (Figure 3a) and more than 40% identity with the putative calcium/calmodulin protein kinase II association domain from *X. campestris* (CaMKII-AD: PDB 3H51). In contrast, alignment with a NTF2 protein family domain (from *Rattus norvegicus*: PDB 1OUN), to which the COG4875 is also predicted as a family member, showed less than 20% identity (Figure 3a). These results indicate that the amino acid sequences of the four domains are highly similar.

The secondary structure of each domain contained in the tpι-CA was predicted using PSIPred webserver [20]. These predictions showed a similar proportion of α-helices (19%), β-strands (32%) and coils (49%) to that from experimentally determined crystal structures from the CaMKII-AD (3H51) and NTF2 (1OUN) protein domains (Table 2). We experimentally confirmed the secondary structural content of tpι-CA using circular dichroism (CD; Figure 3b) and showed that the protein in solution contains 39% of β-strands; 54% coils, which are highly similar to the predicted secondary structure population; and a slightly lower content of α-helices (7%) compared with the predictions.

In order to determine whether the secondary structure of independent COG4875 domains varies within the full-length tpι-CA, we analyzed the secondary structure of several truncated forms of the tpι-CA containing different numbers of domain repetitions by deleting the increasing number of domains from the C-terminus, hereafter referred to as the ∆1-2-3 (composed of domains 1, 2 and 3), ∆1-2 (composed of domains 1 and 2) and ∆1 (composed of domain 1 only) variants. The CD spectrum of each variant (Figure 3b) was analyzed using Dichroweb server [21,22]. The analyses showed that all variants have similar contents of β-strands (31–39%) and unstructured coils (48–55%) in agreement to predictions from PSIPred (Table 2) and that they possess a variable and low content of α-helices (8–19%). This result suggests that the overall secondary structure of the tpι-CA might remain invariable regardless of the number of individual COG4875 domains.

### 2.3. Domain Organization in Tetrameric tpι-CA

The domain repetition in tpι-CA raises the question of their respective organization. We used small-angle X-ray scattering (SAXS) to determine the global structure of the tpι-CA in solution. Size-exclusion chromatography coupled with SAXS on tpι-CA gave rise to a single elution peak, as expected from the abovementioned SEC data (Figure 1b). A Guinier analysis of the X-ray scattering data indicated a radius of gyration of this LMM form of 66.5 ± 0.6 Å and the distance distribution computed from the scattering curve indicated a maximum dimension (D_max_) of 250 Å, suggesting that the protein has a very anisotropic shape. The molecular mass inferred from the data was 292 ± 30 kDa, corresponding to a tetrameric tpι-CA, as observed using MS-ESI (Table 1), cross-linking and SEC (Figure 1b and Figure 2). The global envelope computed from the scattering data is an atypical flat shape with four protruding arms and a ring-like structure in the center. Sixteen copies of the homology models of the COG4875 domains can be accommodated in this global envelope, with eight domains in the doughnut-shaped center of the SAXS envelope and two domains per arm (Figure 4). This global envelope indicates that the four domains are not equivalent and that one moiety constitutes the oligomer interface while the other is exposed to the solvent.

In order to ascertain which moiety was involved in the oligomerization, we measured the hydrodynamic properties of the truncated domain variants presented above. The size exclusion profiles and hydrodynamic radii determined from Diffusion Ordered Spectroscopy-NMR (DOSY-NMR) indicated that these truncated forms remained oligomeric (Figure 5). We thus placed N-terminal domain 1 at the oligomeric interface at the center of the SAXS global envelope and the C-terminal domain 4 in the protruding arms. We then built an atomic model by sequence-based homology modeling using the crystal structure of the *X. campestris* CaMKII-AD (3H51).

The doughnut-shaped center accommodates four copies of the domains 1 and 2. In the X-ray structures of ι-CA domains from *Anabaena* (7C5W and 7C5V) and from *B. natans* (7C5Y and 7C5X), and of the *X. campestris* CaMKII-AD (3H51), the domain-domain interface is composed of two antiparallel β-sheets, and this interface was conserved in our homology model of tpι-CA. This β-sandwich domain-domain interface was observed twice between two domain 1s and twice between two domain 2s. This interface between two domain 1s associates the monomers A and D, and the monomers B and C of the tetrameric tpι-CA (Figure 4). Conversely, the domain 2 pairs do not connect the same monomers: the domain 2 interfaces are between monomers A and B, and monomers C and D. This means that each monomer faces two different chains in its domains 1 and 2. This “turning” or entangled scaffold allows for the tetramerization of tpι-CA (Figure 4b). The short linker between domains 1 and 2 is embedded in the doughnut-shaped center. The SAXS envelope of the arm accommodates domains 3 and 4, together with the linkers between domains 2 and 3 and between domains 3 and 4.

### 2.4. Flexibility in the Protruding Arms Brought by the Linkers

This “drone-like” homology model is coherent with all of the biophysical data and indicates that, despite their high homology in an amino-acid and secondary structure composition, the four domains are not equivalent within the tpι-CA structure. Domains 1 and 2 are involved in dimeric interfaces, while domains 3 and 4 are more exposed to the solvent. The linkers between the domains might be the determining factor controlling this peculiar domain organization. Indeed, the homology between the three linkers (i.e., between domains 1 and 2, domains 2 and 3, and domains 3 and 4) is lower than the homology between domains (Figure 6a), as expected for disordered regions. These linkers are predicted to be flexible linkers using the disorder predictor IU-pred2A [23] and other predictors (Figure 6b) and are expected to bring a high level of flexibility in the tpι-CA arms. The predicted flexibility of the linker 3–4 is higher than that of the other two linkers, as expected from the presence of two proline and three charged residues (Figure 6a). We also calculated the theoretical scattering curve of our atomic model and compared it with the experimental scattering curve using CRYSOL. The fit to the data was fair, with a χ^2^ of 6.2, revealing that the model is good but that there may be some flexibility in the overall architecture of the tetramer, accounting for the slight discrepancies between the two curves in the low-q-region (Figure 4c).

Since the global (and average) SAXS envelope did not account for this putative flexibility, we introduced a flexibility between domains 3 and 4 in our atomic homology model using the program **CO**mplexes with **RA**ndom **L**oops (CORAL) and compared the back-calculated theoretical scattering curves with the experimental SAXS data [24]. Better fits to the data were obtained when flexibility was allowed for this linker compared with rigid models (χ^2^ of 2.1 vs. 6.2, respectively; Figure 4c,d). In the generated structures, the domains 4 were localized in a range of positions that confirm the flexibility of the linkers between domains 3 and 4 (Figure 4b,d). The experimental SAXS data arise from the ensemble of possible conformers, and these average data were best reproduced when the localization of domain 4 was not constrained. This confirms the dynamic nature of the protruding arms of the “drone-like” structure. The presence of a highly flexible linker between domains 3 and 4 might act as a string that competes with a possible domain 4–omain 4 dimerization.

### 2.5. Experimental Validation of the “Drone-Like” Structural Model

In order to validate this domain organization and the atypical “drone-like” shape, we first computed the hydrodynamic radius of the model using HYDROPRO [25], and compared the calculated hydrodynamic properties with the experimentally measured translational diffusion coefficient obtained from DOSY-NMR; they were identical within uncertainty (Figure 5). We further confirmed the position of the domains by analyzing the truncated domain variants mentioned above (Section 2.2). The experimental hydrodynamic properties of the domain-4-deleted construct (∆1-2-3) were identical to that computed from the model in which the domain 4 was deleted from the full-length sequence. The experimental hydrodynamic radius of the two-domain construct (∆1-2) is typical of a spherical tetramer, as expected from the central doughnut shape. The experimental hydrodynamic radius of the domain 1 (∆1) alone is close to that computed from a dimer, as expected from the dimer interface in our model and in other ι-CA domains.

## 3. Discussion

CAs are often cited as a good example of convergent evolution, in which unrelated enzymes evolve to catalyze the same ubiquitous reaction. The different classes forming the CA family are surprisingly diverse in primary, secondary, tertiary and even quaternary structures [11,26]. Here, we studied the features of the overall three-dimensional shape of the recently discovered ι-CA, based on the amino acid sequence of the four repeated domain-containing proteins from *T. pseudonana*. Using a battery of biophysical approaches, we proposed a model of the folding of a homotetrameric tpι-CA in solution, which was also used to infer the structure of the same protein containing fewer domain repetitions (three, two and one).

Based on our results, we confirmed that the previously described LMM form [6] corresponds to a stable tetrameric form in solution. Due to the characteristic subcellular localization of the ι-CA towards the periphery of the plastid of photosynthetic eukaryotes [6,15], it is unlikely that the HMM-nucleic acid form occurs in vivo and, thus, could be an unspecific artefactual association of multiple ι-CA monomers together with nucleic acids. However, the possibility that the ι-CA could interact with other cellular components (e.g., other proteins and lipids) cannot be discarded, in particular because both LMM and HMM are active and catalyze CO_2_ protonation [6]. The nucleic acid-bound HMM might mimic other forms induced by interaction with other negatively charged surfaces, such as galactolipids [27] that are abundant in plastid membranes. Such high molecular weight forms of CA with undetermined mass were also observed for other CAs such as a stromal β-CA, PtCA1, from the diatom *Phaeodactylum tricornutum*, which can form aggregated structures when purified [28]. Its association within large clumped macromolecular complexes was confirmed in vivo, as was also shown for the homologous PtCA2 [29]. This complex formation was possible through a C-terminal amphipathic α-helix exposed to the solvent that does not participate in the dimerization of the PtCA [29] and is also present in other β-CAs from other diatom species [30]. Interactants of the PtCA1 and PtCA2 have not yet been found, but an interaction with lipids (e.g., galactolipids) or carbohydrates has been hypothesized [29]. In this same context, further studies are necessary to show whether the tpι-CA is also able to form a complex with other cell components and which structural feature would allow this.

The ESI-MS, SEC and SAXS data indicated that the LMM form of tpι-CA is a tetramer. This is in contrast with the multi-domain ι-CA from the chlorarachniophyte alga *B. natans* that forms a homodimer [15]. The SAXS and DOSY-NMR data indicated that the global envelope has an atypical “drone-like” shape with a central core (domains 1 and 2) and four protruding arms (domains 3 and 4), and this also differs from the X-ray proposed structure of *B natans* ι-CA, which is an elongated dimer [15] (Figure 7).

The CD data indicated that the secondary structure composition of each of the four tpι-CA domains is similar to other COG4875 domains, as expected from the high sequence conservation within this class. The secondary structure composition is also very close to the COG4337 domains, which are constituents of a metal-free ι-CA from *B. natans* and the cyanobacterium *Anabaena sp.* PCC7120. The high sequence identity with COG4875 domains for which X-ray structures are available enabled us to build a homology atomic model that we constrained within the SAXS average envelope. In the three COG4337 domains of ι-CA from *B. natans*, the active site for the CO_2_ protonation was composed of the residues Thr159/322/486, Tyr176/339/503, His256/420/584 and Ser258/422/586, with the last two being part of the specific HHHSS sequence and which are all oriented to the core of the domains [15]. In all tpι-CA domains, all these residues are present (Thr62/191/320/448, Tyr77/206/335/463, His139/268/397/525 and Ser141/270/399/527, Figure 7) and oriented towards the core of the domain. Notably, both *B. natans* ι-CA and tpι-CA are inhibited by Zn^2+^, and this peculiar property might be specific to these catalytic residues. Moreover, as in the ι-CA from *B. natans*, in tpι-CA, CO_2_ can be protonated even in the absence of metal ion. In CAs, CO_2_ is proposed to be positioned near phenylalanine residues. Phe177/340/504 and Phe193/357/521 in *B. natans* ι-CA are proximal to the active site and are conserved in all four tpι-CA domains (Phe78/207/336/464 and Phe81/210/339/467, Figure 7). The CA activity of the four-domain tpι-CA as well as the variant constructs ∆1-2-3 and ∆1-2, has been previously confirmed [6]; however, whether all domains contribute to the overall CA activity or to metal binding in a particular tetrameric conformation is still unknown and must be further investigated.

We validated the localization of the domains in our model by comparing the predicted hydrodynamic properties of the FL and domain truncated variants with experimentally measured diffusion coefficients by DOSY-NMR. Back calculation of the SAXS curve from the model fitted the experimental data better when the C-terminal domain 4 localization was unconstrained, indicating that the protruding arms are flexible, and this flexibility might be conferred by the linker between domains 3 and 4 that was predicted to be a long-disordered linker by IUPred2A. This particular linker is also predicted to be disordered in the ι-CA sequences from other diatom species with four-domains, including *Cyclotella cryptica*, *Fistulifera solaris* and *Thalassiosira oceanica*. However, it is absent in the C-terminal linkers from homologous sequences having less domain repeats (data not shown). This suggests that the protruding and flexible arms observed in the SAXS envelope from the tetrameric tpι-CA is a particular feature of the four-domain ι-CA and, in agreement with our proposed models (Figure 5b), does not exist in other homologous sequences with fewer domains.

Tpι-CA domains 1 and 2 are associated in a dimer with their β-sheet surface, which includes the specific HHHSS sequence [14], embedded in a β-sandwich interface. This interface also contains conserved Arg and Cys residues (Arg122/251/380/508 and Cys105/231). Interestingly, only domains 1 and 2 possess the cysteine residues that might stabilize the dimer interface. The distance between the His269 residues in the domain 2–domain 2 β-sandwich is less than 4.5 Å and similar to what was found between the His257 residues in *B. natans* ι-CA (Figure 8). On the contrary, in the domain 1–domain 1 β-sandwich, the distance between the His140 residues is more than 7 Å (Figure 8). This larger interface also includes highly conserved Asp and Glu residues from the neighboring domains 2 (Asp259 and Glu260, Figure 8) and the His139 and His140 residues that are the homologues of the residues that were predicted to coordinate metal [6,14]. In the present dataset of Mn-bound proteins, metal coordination involved mainly His, Glu and Asp residues [31,32]. We hypothesize that the pair of His140 (domain 1), pair of Asp259 and pair of Glu260 (domain 2) residues that are all localized within 10 Å in our homology model might be involved in Mn^2+^ coordination, and this can be further tested by site-directed mutagenesis.

In our model, the domain 4s do not interact and the β-sheet surface that composes the β-sandwich domain-dimerization interface in the other COG4875 domains is protected by the C-terminal extension (Figure 9). This C-terminal extension has a peculiar amino-acid composition with a high number of hydrophobic residues (oriented toward the β-sheet surface) surrounded by glutamic acid residues exposed to the solvent. The C-terminal extension might act as a “gate” that prevents domain 4-domain 4 dimerization.

## 4. Conclusion

A comparison of the tpι-CA structure with other existing CA structures revealed that, while there are elements of the fold that resemble previously known structures, the overall fold is novel as was anticipated from the absence of sequence conservation. The tpι-CA monomer has an unusual four domain repeat with a non-compact appearance. The tetramer has a drone-shape, comprising a doughnut core flanked by four protruding extensions mediated by the flexibility of the last linker. Its atypical shape might be linked to its localization in the appressed intermembrane space of the chloroplast endoplasmic reticulum (CER) [2]. It is intriguing to speculate that the drone structure of tpι-CA may be linked to its CO_2_/HCO_3_^−^ delivery function as is the case for delivery by human-made drones.

## 5. Materials and Methods

### 5.1. Protein Expression and Purification

The DNA sequence of the four-domain-containing ι-CA from *T. pseudonana* was produced synthetically (GeneCust, Ellange, Luxembourg) based on the cDNA sequence without the nucleotides coding for the signal peptide and inserted between the NdeI and XhoI restriction sites of a pET-28a+vector so that the protein was fused to a His-tag on its N-terminus. The same procedure was performed for the ι-CA variants containing three, two and one domains, always removing domains from the C-terminal extremity. The resulting vectors containing either the ι-CA gene or its domain variants were cloned in the *Escherichia coli* strain BL21-C41(DE3). The expression of recombinant proteins in *E. coli* was induced by 1 mM Isopropyl β-D-1-thiogalactopyranoside (IPTG) at 37 °C for 5 h. Cell pellets were resuspended in a buffer containing 50 mM sodium phosphate, 10 mM imidazole and 50 mM NaCl buffer (pH 8), plus lysozyme and protease inhibitor cocktail. Cells were broken by sonication. Lysates were centrifuged for 30 min at 16,000 g and 4 °C, and the supernatant was loaded onto a Ni-NTA column (height 6 cm and diameter 1.5 cm). The column was washed with a buffer containing 0.15 M imidazole. Elution of the ι-CA was performed with a buffer containing 0.35 M imidazole.

### 5.2. Size Exclusion Chromatography (SEC)

SEC chromatography was performed on a Superdex Increase S200 10 × 300 (mm × mm) at 4 °C. Five hundred microliters of the sample were loaded at varying concentrations ranging from 50 µM to 200 µM. The elution volumes were monitored by absorbance at 215 nm and 280 nm. When mentioned, 25 U mL^−1^ of benzonase (Sigma) with 1 mM MgCl_2_ was added to the sample, and nucleic acid digestion was performed for 1 h at 37 °C followed by an overnight incubation at 4 °C. When mentioned, 3 U µL^−1^ of thrombin was added to the sample, and proteolysis of the His-tag was performed by incubation overnight at 4 °C.

### 5.3. Congo Red Assay

A solution of 1 mM Congo red (CR) was prepared in a buffer of 20 mM Tris and 50 mM NaCl (pH 8). A mix of 70 µM of Congo red and 10 µM of the protein sample was prepared, and the absorption spectrum between 300 and 700 nm wavelengths was recorded using a Perkin Elmer Lambda 25 UV/Vis spectrophotometer. The spectrum of the protein solution without CR was subtracted to the spectrum of the protein–CR mix. The spectrum of the CR solution alone was also recorded for comparison. A control of fibril formation was performed using lysozymes heated at 55 °C for 5 min, as described previously [33].

### 5.4. Protein Cross-Linking

One hundred micrograms (100 µg) of purified protein extract in 20 mM Na_2_HPO_4_ buffer (pH 8.0) was mixed with 0.01% glutaraldehyde. The mixture was then incubated for 10 to 15 min at room temperature, and then, the reaction was stopped by the addition of 80 mM Tris-HCl (pH 8.0). The reaction was immediately mixed with a Laemmli buffer for further SDS-PAGE and Western blot analysis.

### 5.5. Protein Analysis

Protein samples from the purified protein were mixed with the sample buffer (62.5 mM Tris, 2.5% SDS, 0.002% bromophenol blue, 10% glycerol, 20 mM DTT and pH 6.8) and denatured by heating at 85 °C for 5–10 min. Samples of 20 to 30 μg protein extracts or 2 to 5 μg of purified proteins were loaded onto 12% acrylamide/bis-acrylamide gels and run at 120 volts until the migration front reached the bottom of the gel. Electrophoresis was performed in a Bio-Rad Mini Protean III system (Bio-Rad, Hercules, California, United States) using a buffer of 50 mM Tris, 380 mM glycine and 10% SDS.

After electrophoresis, the gel was either stained with Coomassie blue or used for Western blot analysis. For Western blot, the proteins were transferred to a nitrocellulose membrane (0.2 μm; Carl Roth Gmbh, Karlsruhe, Germany) with active transfer at 80 volts for 1 h. The transfer buffer contained 25 mM Tris, 121 mM glycine and 20% ethanol. After transfer, the membrane was blocked with a solution of 5% low fat milk in TBS-T (50 mM Tris, 150 mM NaCl, 0.05% Tween-20 and pH 7.6). The membrane was then incubated for 1 h at room temperature or overnight at 7 °C with the primary antibody (1:1000 dilution). The secondary antibody (anti-rabbit IgG horseradish peroxidase, 1:10,000 dilution) was incubated for 1 h at room temperature. The membrane was revealed with the Enhanced Chemiluminescence technique and then visualized with a digital imaging system (ImageQuant LAS 4000 mini, GE, Chicago, Illinois, United States).

### 5.6. Analytical Ultracentrifugation (AUC)

Sedimentation velocity experiments were carried out at 40,000 rpm and 20 °C in a Beckman Optima-XL-A analytical ultracentrifuge using 1.2 cm or 0.3 cm double sector centerpieces in an AN50Ti rotor. Scans were acquired in the continuous mode at 280 nm in the range of 0.1 to 1 absorption. All ι-CA samples were in a 20 mM Tris and 50 mM NaCl (pH 8) buffer. At 20 °C, the partial specific volume of ι-CA, the solvent density and the viscosity calculated with SEDNTERP (jphilo. Available online: http://www.jphilo.mailway.com/index.htm (accessed on 10th December 2018)) [34] were 0.734501 mL g^−1^, 1.0009 g cm^−3^ and 0.01002 poise, respectively. The data recorded from moving boundaries were analyzed in terms of continuous size distribution functions of sedimentation coefficient, C(S), using the program SEDFIT [35]. The sedimentation coefficient was measured at different protein concentrations, and the standard sedimentation coefficient was obtained by extrapolation to a concentration of the protein equal to zero. All S values of ι-CA were corrected to standard conditions—i.e., 20 °C in water—by SEDFIT.

### 5.7. Electrospray Mass Spectrometry (ESI-MS)

Purified recombinant tpι-CA in 20 mM Tris-HCl, 50 mM NaCl, and pH 8 was buffer-exchanged using a micro Bio-Spin column Bio-Gel P6 (Bio-Rad) against a 500 mM aqueous ammonium acetate pH 8 solution for native mass spectrometry (MS) at a final protein concentration of 8 µM measured by a Thermo Scientific nanodrop 2000 C (at lambda 280 nm, epsilon 110.356 µM^−1^ cm^−1^). The MS parameters used in the electrospray Q-ToF mass spectrometer (Synapt G1, Waters) were set as source temperature 20 °C, capillary voltage 1.5 kV, sampling cone 140, extraction cone 4, trap collision energy 40, transfer collision energy 30 and *m*/*z* window 6,000 to 10,000 to detect the oligomers of tpι-CA. The neutral molecular mass was manually calculated from spectra by averaging adjacent *m*/*z* from five consecutive charge-state assigned peaks. The deduced molecular mass was compared with the theoretical mass of ι-CA, which has been deduced from the sequence including the His-tag (GSSHHHHHHSSGLV…: MW = 64,988 Da), as confirmed by N-ter sequencing (data not shown).

### 5.8. Circular Dichroism (CD)

The purified recombinant protein was prepared at 2 µM in filtered 20 mM Na_2_HPO_4_ buffer (pH 8). The circular dichroism (CD) spectra were recorded from 260 to 180 nm in a Jasco J-815 CD Spectrometer (JASCO. Easton, Maryland, United States) at 25 °C in a 2 mm path quartz cuvette. The raw values of ellipticity (mdeg) were converted into mean residue molar ellipticity (θ) using the following formula:$$\theta \left(\deg cm^{2}dmol^{-1}res^{-1}\right)=\frac{MRW\;\times \;E\left(\deg\right)}{d\;\times \;c\;\times \;10}$$
where “*MRW*” corresponds to the mean residue weight, “*E*” is the raw ellipticity, “*d*” is the path length of the cuvette (cm) and “*c*” is the sample concentration (g cm^−3^). The contents of α-sheets and β-helices were estimated using DichroWeb software [22].

### 5.9. Dynamic Light Scattering (DLS)

Purified ι-CA was analyzed by DLS using a Zetasizer Nano ZS (Malvern Instruments, Malvern, United Kingdom). The proteins samples prepared at 3 to 5 mg mL^−1^ in a buffer of 20 mM Tris and 50 mM NaCl (pH 8) were centrifuged for 15 min at 14,000 rpm at 4 °C prior to DLS measurement. Three measurements consisting of 10 runs, each 10 s, were performed on ι-CA sample with a scatter angle of 173 degrees.

### 5.10. Size Exclusion Chromatography Coupled to Small-Angle X-Ray Scattering (SEC-SAXS)

The SEC-SAXS experiments were performed at the SWING beamline of SOLEIL Synchrotron (Gif-Sur-Yvette, France). A HPLC column Agilent Bio SEC-5, 500 Å (length: 300 mm, particle size: 5 µm) was used prior to SAXS measurements. All experiments were performed at 15 °C. The sample-to-detector (Eiger 4M) distance was adjusted to 2 m, giving access to scattering vector q = 4π/λ·sinθ (where 2θ is the scattering angle and λ is the wavelength, equal to 1.033 Å) ranging from 0.011 to 0.50 Å^−1^. Two-hundred frames of 990 ms with 10 ms dead-time were recorded during the first minutes of the elution for the background signal. The signal of the protein was recorded during all protein elutions.

The raw data were processed using the dedicated in-house software Foxtrot. The buffer signal was subtracted, and careful inspection of the protein scattering data allowed us to average the identical scattering curves recorded during protein elution. Data analysis was performed using the ATSAS suite of [36]. The Rg was obtained via PRIMUS using the Guinier approximation, and the distance distribution function P(r) was obtained via GNOM. The molecular mass was assessed using SAXS-MoW [37]. Ab initio 3D models corresponding to the scattering envelopes were calculated using DAMMIF [24] with P4 symmetry, and using GASBOR with the number of amino acids of the protein monomer as input and with P4 symmetry [24]. The atomic models were assessed and refined using CRYSOL [36] and CORAL [24]. CORAL started with the atomic model as input and was allowed to move the domains 4 independently and to build new CA traces between the domains 3 and 4. The SAXS data have been deposited at SAS BDB (draft ID 3391).

### 5.11. Diffusion-Ordered Spectroscopy Nuclear Magnetic Resonance (DOSY-NMR)

DOSY-NMR was performed on the purified ι-CA and its domain variants. A 500 µL sample was prepared at 30 to 80 µM in a buffer with 27 mM Tris, 45 mM NaCl, 10% D_2_O and 1 µL DSS (4,4-dimethyl-4-silapentane-1-sulfonic acid) at pH 8. The measurements were repeated with 10 increasing strengths for a duration of 2.8 ms. A time delay of 200 ms was used to allow the molecules to diffuse before gradient decoding and diffusion. The experiments were performed at 298 K on a 600 MHz Bruker advance II spectrometer equipped with a cryo-probe. The spectra were transformed by NMRPipe [38], and the diffusion coefficients were calculated using Octave (GNU Octave. Available online: https://www.gnu.org/software/octave/ (accessed on 18th December 2018)). The diffusion coefficient (*D_t_*) obtained was used to calculate the hydrodynamic radius (*R_h_*) of each sample using the Einstein–Stokes equation:$$D_{t}=\frac{k_{B}T}{6\pi \eta R_{h}}$$
where *k_B_* is the Boltzmann constant (1.380 × 10^−23^ kg m^2^ s^−2^ K^−1^), *T* is the temperature (in Kelvin) and *η* is the viscosity of the medium.

The translational diffusion coefficient of the structural models was computed using HYDROPRO version 10 [25]. Standard parameters were used for the HYDROPRO calculation; the radius of primary elements was set to 2.9 Å, six beads sizes were used, which ranged from 10 to 20 Å for the full-length construct, from 7 to 14 Å for the Δ123 construct, from 5 to 10 Å for the Δ12 construct and from 2 to 4 Å for the Δ1 construct; the temperature was set to 293 K to match the experimental diffusion measurements; and the viscosity of the solvent was set to 0.01 poises.

### 5.12. D-Modeling

Homodimer models of domains 1, 2 and 3 were built using the homology modelling server SWISS-MODEL with the homodimer structure of the putative Calcium/Calmoduline-dependent kinase II association domain from *Xanthomonas campestri* (3H51.pdb) as a template [39]. The non-dimeric C-terminal domain 4, where a mostly hydrophobic C-terminal extension covers the monomer interface, which is responsible for the dimerization of the other domains, was built using the I-Tasser server [40]. The homodimer domains 1, 2 and 3 and the monomeric C-terminal domain were assembled manually in the SAXS envelope using the PyMol program. In this step, individual N- and C-terminals of each domain were carefully oriented to allow for their connection to the full-length monomer, constraining their position in the overall structure. Linker regions between domains were added and minimized, and some colliding loops were moved using the Wincoot structural modeling program [41]. The overall geometry of a connected 4 domain monomer was refined using the ModRefiner program [42] and the tetrameric assembly reconstructed with this refined model. Finally, the whole tetrameric assembly was refined using the SWISS-MODEL server with the tetrameric model from the previous step as the starting template.

## Figures and Tables

**Figure 1 ijms-22-08723-f001:**
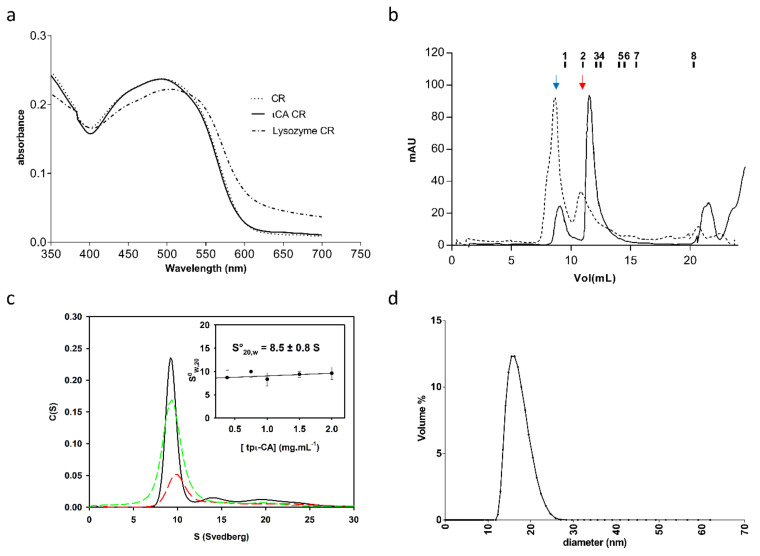
Oligomerization state of the tpι-CA. (**a**) Congo red (CR) spectral shift assay from purified tpι-CA. The spectrum of CR alone and mixed with a tpι-CA sample is shown. The presence of fibrils is shown by a shift in the spectrum of a CR-lysozyme control previously heated at 55 °C for 5 min prior to assay. (**b**) Size exclusion chromatography of recombinant tpι-CA. The HMW form (dotted line) produces the LMM form (plain line) upon treatment with Benzonase. The elution volume of the LMM form is 10.71 mL which corresponds to an apparent MW of 280 kDa. The elution volumes of standard proteins are indicated above the profile: 1—blue dextran, 2—ferritin (440 kDa), 3—catalase (240 kDa), 4—aldolase (158 kDa), 5—Bovine Serum Albumin (BSA) dimer (136 kDa), 6—BSA monomer (68 kDa), 7—ovalbumin (43 kDa) and 8—Cytochrome C (12.5 kDa). Blue and red arrows show peaks corresponding to the HMM and LMM, respectively. (**c**) Sedimentation velocity experiment in an analytical ultracentrifugation (AUC) performed on the purified LMM form at different concentrations: 2.0 (black), 1.5 (green dashed) and 0.7 (red dashed) mg mL^−1^. Standard sedimentation coefficient S^0^_20,W_ determination is obtained by extrapolating the S_20,W_ value at protein concentration equal to zero, as shown in the inset. (**d**) Dynamic light scattering (DLS) curve of the LMM form.

**Figure 2 ijms-22-08723-f002:**
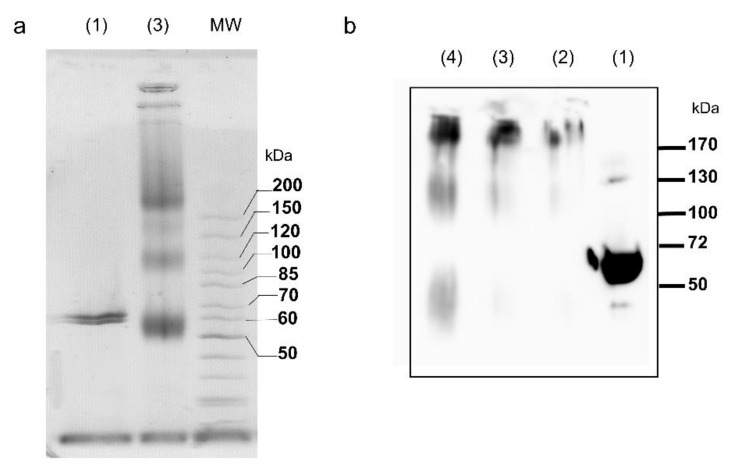
Glutaraldehyde-induced protein cross-linking. (**a**) SDS-PAGE and (**b**) Western blot of the following samples: (1) untreated purified tpι-CA, 5 µg; (2–4) cross-linked purified tpι-CA, 2, 5 and 10 µg, respectively. MW: molecular weight markers.

**Figure 3 ijms-22-08723-f003:**
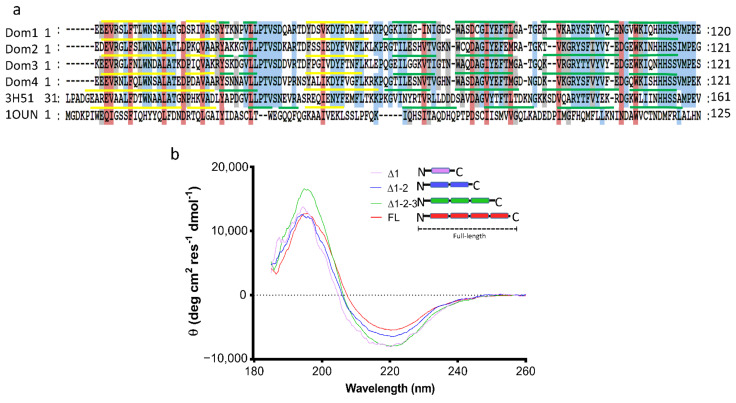
Analysis of the secondary structure. (**a**) Alignment of the amino acid sequence of each domain contained in the full-length tpι-CA; 3H51 and 1OUN correspond to the PDB sequences of the CaMKII-AD from *Xanthomonas campestris* and the NTF2 from *Rattus norvegicus*, respectively. Alignments were performed using MEGA4 software and analyzed using GeneDoc (University of Pittsburg. Available online: http://www.psc.edu/biomed/genedoc (accessed on 15th June 2021)). Shading levels correspond to the conservation of amino acids: Red, above 80% of identity; blue, 70% identity; and light grey, 60% identity. The yellow and green lines above each sequence represent the presences of α-helices and β-strands, respectively. The secondary structures of the four tpι-CA domains were predicted using PSIPred software; for 3H51 and 1OUN, the secondary structures were obtained from their crystal structures. (**b**) Circular dichroism spectra of full-length tpι-CA and of the variants containing one, two or three domain repetitions.

**Figure 4 ijms-22-08723-f004:**
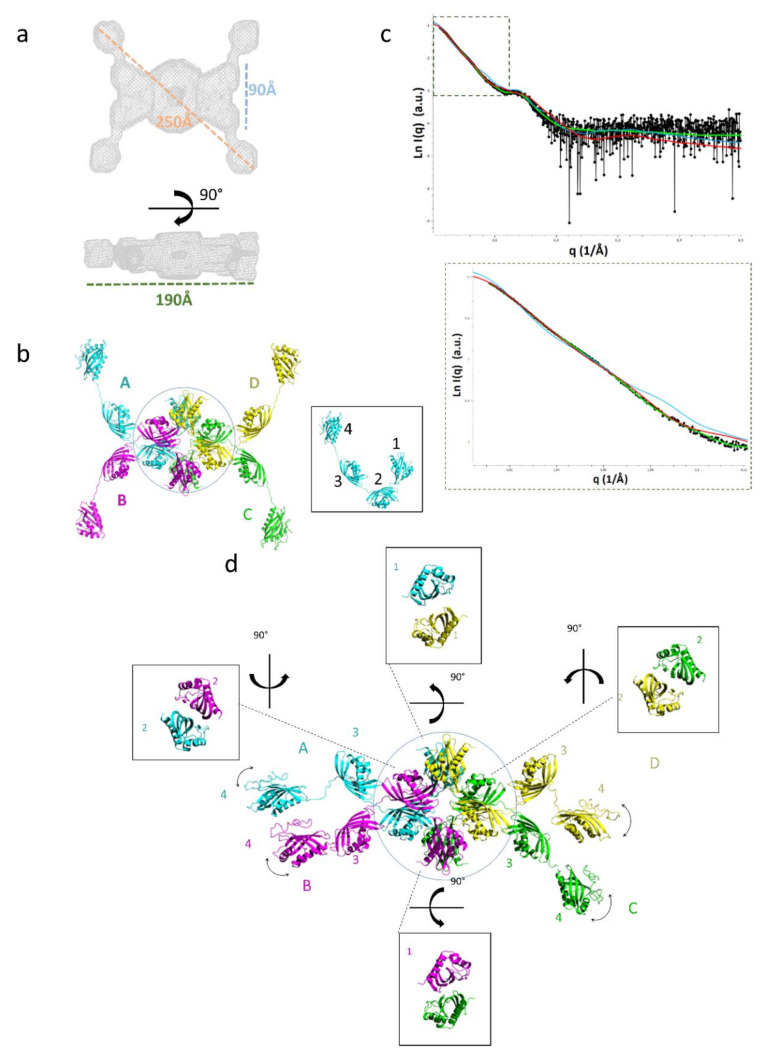
Three-dimensional modeling of the tetrameric full-length tpι-CA. (**a**) Ab initio SAXS-based model envelope inferred from the experimental data. *Top*: front view; *bottom*: lateral view. (**b**) Proposed model of the tetramer based on a SAXS envelope. Colored letters are used to designate each monomer. The box on the right shows the orientation of each domain within one monomer. (**c**) *Top*: The experimental scattering curve (black circles) is compared with the calculated scattering curves of the shape determined by DAMMIF (green curve, χ^2^ of 2.1), of the initial atomic model (blue curve, χ^2^ of 6.2) and of a model generated by CORAL (red curve, χ^2^ of 2.1); *bottom*: A zoom on the data at low q (dashed box) is shown. (**d**) The CORAL model is represented in ribbons, with one color per monomeric chain, as in (**b**). The interface of each of the domain pairs contained in the central core (circled) is shown.

**Figure 5 ijms-22-08723-f005:**
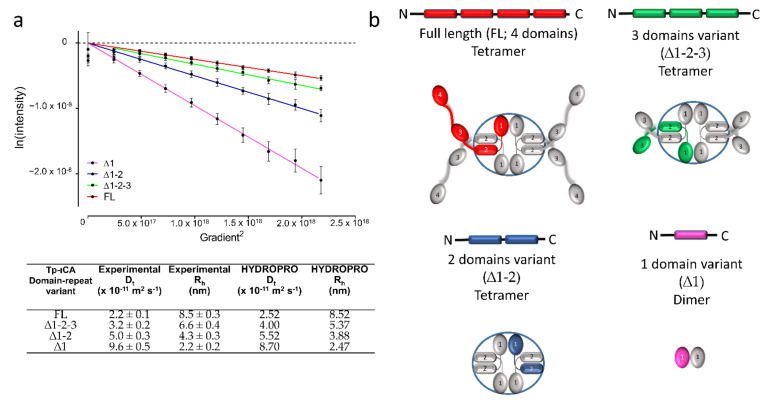
Domain organization of tpι-CA. (**a**) Translational diffusion coefficient (D_t_) of the different domain variants produced from the full-length (four domain-containing) tpι-CA obtained from DOSY-NMR. *Top*: Logarithm of the NMR signal intensity as a function of the square of the gradient strength. *Bottom*: Table showing experimental D_t_ for the inferred R_h_ using the Stokes–Einstein relation as well as the computed D_t_ and R_h_ from the homology model using HYDROPRO software. The calculation for the full-length construct and the domain variant constructs ∆1-2-3 and ∆1-2 are performed on the tetrameric forms. The calculation of the domain variant ∆1 is performed on the dimeric form. (**b**) Schematic models of the different tpι-CA domain variants from which the HYDROPRO calculation was performed. Only one monomer was colored in each model.

**Figure 6 ijms-22-08723-f006:**
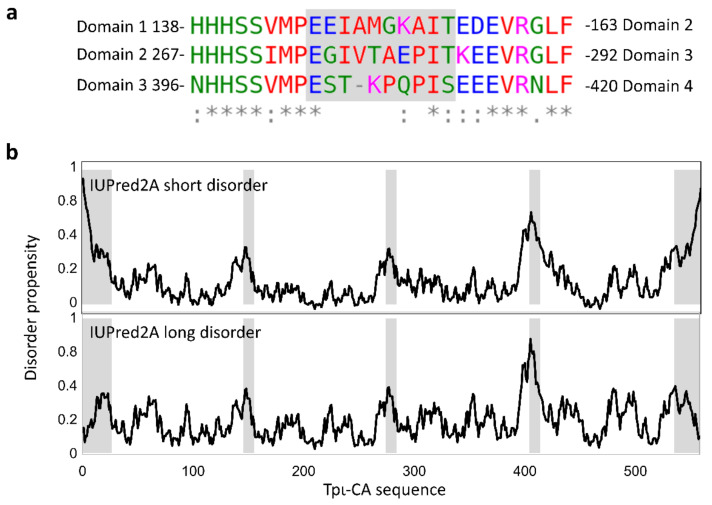
Flexibility of the interdomain linkers. (**a**) Clustalω alignments of the linkers between domains 1 and 2, domains 2 and 3, and domains 3 and 4. The linker residues are shaded in grey. (**b**) IUPred2A disorder prediction for short disordered regions (top) or long disordered regions (bottom). Only the linker between domains 3 and 4 is predicted to be a long-disordered region.

**Figure 7 ijms-22-08723-f007:**
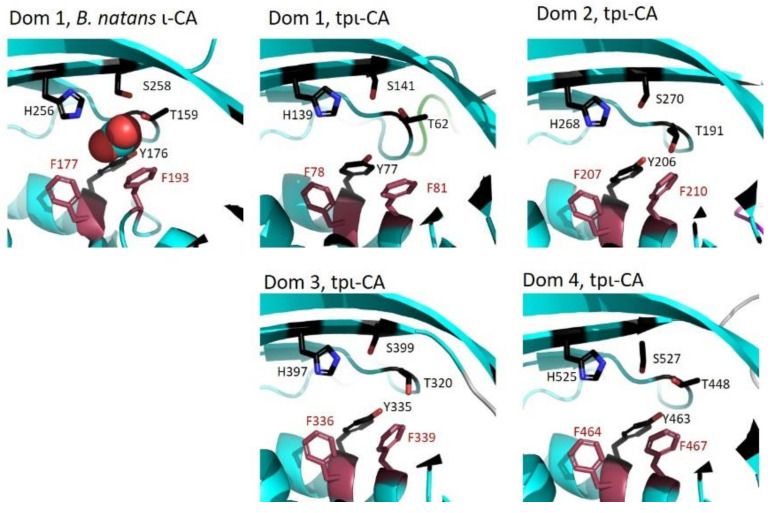
Structure of the ι-CA active site in domain 1 of *B. natans* ι-CA (pdb 7C5X and of each of the four domains of tpι-CA (homology model).

**Figure 8 ijms-22-08723-f008:**
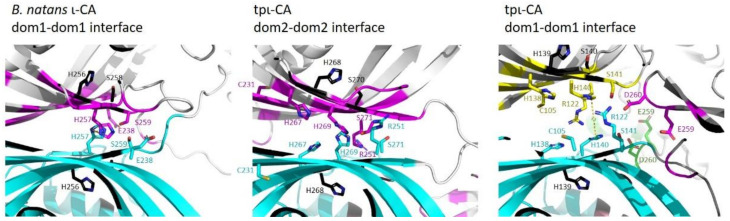
Structure of the ι-CA β-sandwich interface between the domain 1s of *B. natans* ι-CA (pdb 7C5X), between domains 2 and 1 of tpι-CA. The monomer A is colored blue, and the residues within 5 Å of the monomer A are colored magenta for those of monomer B, green for those of monomer C and yellow for those of monomer D. The distance between His257 residues in *B. natans* ι-CA is 4.5 Å, between His269 residues in the domains 2 of tpι-CA is 3.5 Å and between His 140 in the domains 1 of tpι-CA is 7.1 Å.

**Figure 9 ijms-22-08723-f009:**
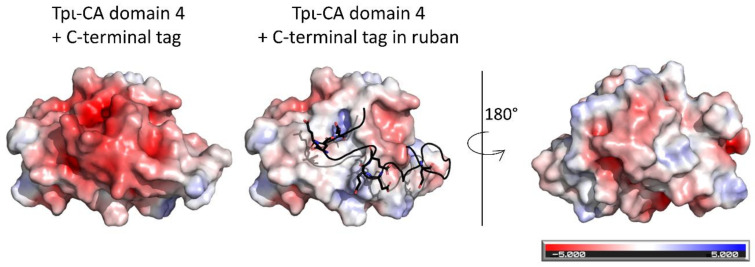
Electrostatic surface of the tpι-CA domain 4 with the C-terminal extension (**left**) or without (**right**). The C-terminal extension is represented in cartoon with the Glu residues in stick. The color of the surface corresponds to the electrostatic charge, which ranges from −5 (red) to +5 (blue).

**Table 1 ijms-22-08723-t001:** Molecular mass and oligomeric state of the full-length tpι-CA determined by ESI-MS.

Charge State	*m*/*z*		Delta *m*/*z*
	**Theoretical**	**Experimental ^*^**	
28	9285.00	9288.00	3.00
29	8964.86	8970.22	5.36
30	8666.07	8670.32	4.25
31	8386.55	8390.12	3.57
32	8124.50	8127.45	2.95
**Deduced multimeric mass (Da)**	**Oligomer state**	**Deduced monomer mass** **(theoretical) (Da)**	**Error (ppm)**
260,066.20	4	65,016.55 (64,988)	439

* As an example, an *m*/*z* of 9288 with a charge state of 28 gives a ((9288 × 28) − 28) or 260,036 Da molecular mass.

**Table 2 ijms-22-08723-t002:** Proportion of secondary structural elements in tpι-CA, derived either from the experimental CD data or from prediction.

Tp-ιCA Domain-Repeat Variant	Experimental CD (DichroWeb Analysis)			Predicted Secondary Structure (PSIPred)		
	β-Strand	α-Helix	Coil	β-Strand	α-Helix	Coil
FL	0.39	0.07	0.54	0.32	0.19	0.49
∆1-2-3	0.35	0.18	0.48	0.33	0.17	0.50
∆1-2	0.37	0.08	0.55	0.34	0.17	0.49
∆1	0.31	0.19	0.50	0.32	0.16	0.52

## Data Availability

Not applicable.

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
