# Peer review of "Structural Contour Map of the Iota Carbonic Anhydrase from the Diatom Thalassiosira pseudonana Using a Multiprong Approach"

_ijms, 2021, doi:10.3390/ijms22168723_

Round 1

Reviewer 1 Report

The manuscript describes the biophysical characterization of the iota class carbonic anhydrase from T. pseudonana (Tp-CA).  It employs a series of techniques (SEC, MALS, SAXS, NMR, ESI-MS, CD, etc.) to interrogate the quaternary structure of the protein in the absence of an explicit crystal structure. The Tp-CA protein contains 4 domains which share 60% sequence identity. The authors model the Tp-CA structure based on homology models and propose a “drone-like” quaternary structure of the protein wherein domains 1 and 2 form a donut-like central core, and domains 3 and 4 form the outer “wings” of the molecule; domain 4 is the most solvent exposed domain. The authors not that the protein forms the tetramer when domains 1 & 2 are present; domain 1 dimerizes in solution.

     The manuscript is well constructed, laid out and easily followed by the reader, and the data well represented and displayed. I only have minor comments, mostly of a typographical nature (below). My main question for the authors is the deposition status of the SAXS data. The authors do not mention in Section 4.11 (Page 15) where this data is deposited. Is the data in either the BIOISIS or SASDB?  They are strongly encouraged to submit to either of these databases if they have not already done so, and of course to reference that deposition if they already have.

Minor issues.

  1. The authors did not properly italicize bacterial names it the 3rd paragraph of the introduction (page 2, line6-76); they are done elsewhere, so a simple fix.
  2. Some of the numerical data is presented with a decimal point while others use a comma, for instance Table 1; please format for consistency.
  3. Pg 3, line 99 – it should read “…Benzonase – a nuclease …”. Please add and “a”
  4. Pg 3, Line 116, remove the period between “also.in”
  5. Several times in the text “Table”, “Figure” or “Section” are left uncapitalized, such as Pg. 5, Line 152. Please fix.
  6. Page 13, Line 427 (Section 4.4) – It should read “One hundred micrograms…”

Author Response

We thank the reviewer for all the suggestions and observations provided to improve our manuscript. We have checked point by point all reviewer’s comments and made the corresponding modifications to the main text. The English has now been checked by a native English speaker.

The reviewer wrote:

“The manuscript describes the biophysical characterization of the iota class carbonic anhydrase from T. pseudonana (Tp-CA).  It employs a series of techniques (SEC, MALS, SAXS, NMR, ESI-MS, CD, etc.) to interrogate the quaternary structure of the protein in the absence of an explicit crystal structure. The Tp-CA protein contains 4 domains which share 60% sequence identity. The authors model the Tp-CA structure based on homology models and propose a “drone-like” quaternary structure of the protein wherein domains 1 and 2 form a donut-like central core, and domains 3 and 4 form the outer “wings” of the molecule; domain 4 is the most solvent exposed domain. The authors not that the protein forms the tetramer when domains 1 & 2 are present; domain 1 dimerizes in solution.

     The manuscript is well constructed, laid out and easily followed by the reader, and the data well represented and displayed. I only have minor comments, mostly of a typographical nature (below). My main question for the authors is the deposition status of the SAXS data. The authors do not mention in Section 4.11 (Page 15) where this data is deposited. Is the data in either the BIOISIS or SASDB?  They are strongly encouraged to submit to either of these databases if they have not already done so, and of course to reference that deposition if they already have.”

Reply: We thank the reviewer for this suggestion. SAXS data has been recently deposited at SASBDB and we are currently waiting for its revision and submission (draft ID 3391).  We added this to the main text, section 4.11, page 15, line 525.

Minor issues.

  1. The authors did not properly italicize bacterial names it the 3rd paragraph of the introduction (page 2, line6-76); they are done elsewhere, so a simple fix.

Reply: This might be a mistake after first edition, as our submitted version did have all scientific names in italics. We corrected them in the introduction and throughout the whole text 

  1. Some of the numerical data is presented with a decimal point while others use a comma, for instance Table 1; please format for consistency.

Reply: This was corrected

  1. Pg 3, line 99 – it should read “…Benzonase – a nuclease …”. Please add and “a”

Reply: This was corrected

  1. Pg 3, Line 116, remove the period between “also.in” 

Reply: This was corrected

  1. Several times in the text “Table”, “Figure” or “Section” are left uncapitalized, such as Pg. 5, Line 152. Please fix.

Reply: This was corrected and checked throughout the text. 

  1. Page 13, Line 427 (Section 4.4) – It should read “One hundred micrograms…”

Reply: This was corrected

Reviewer 2 Report

Review of the manuscript by Jensen et.al. “Structural contour map of the iota carbonic anhydrase from the 2 diatom Thalassiosira pseudonana, using a multiprong approach”.

Number of Carbonic anhydrases (CAs) studies is huge and any report on another new member of the family should extend our knowledge specifically. CAs are a family of enzymes that catalyze the interconversion of CO2 to HCO3- and they are ubiquitous. Authors described the “iota”class first found in the marine diatom Thalassiosira pseudonana and widespread among photosynthetic microalgae and prokaryotes. The i-15CA is composed of  a domain COG4875 (or COG4337) that can be repeated from one to several times, and shows resemblance to a calcium-calmodulin protein kinase II association domain (CaMKII-AD). The crystal structure of this domain from a cyanobacterium and a chlorarachniophyte has been recently determined.

Numerous experimental techniques and methods were used to characterize the enzyme which seems to have rather complex structure and behavior. I think that major achievement of the paper is careful analysis of the SAXS provided structures and fitting of all CAs molecules (each composed of 4 domains)  into the low resolution structure.

There are nevertheless some issues which I would like to see addressed or at least discussed.

  • The existence of HMM and LMM forms seems to be caused by the presence of either DNA or RNA. How the formation of CA and nucleic acids complexes is possible and why CA would bind (non specifically) nucleic acids
  • All methods finally lead to reconstruction of 3D structure of the tetrameric complex based on secondary structure prediction, disorder prediction and homology modelling. Despite the fact that interfaces between domains of individual CAs in the complex were modelled and are described, there is no adjustment of interfaces studied by Molecular Dynamics or other simulation approaches. Did authors perform or do they think about a computational analysis of the final composition?
  • The most important question is if the activity of the CA depends on the presence of all 4 domains composing tetra-domain protein. Did authors addressed such question or do they have any experimental result clarifying roles of particular domain for the function?
  • It is clear that there is no homology between linkers connecting individual domains. On the other hand authors could try to find homologues of linkers over all know protein sequences to find if this connectivity element is unique or used in similar cases. Please comment it.

Author Response

We thank the reviewer for all the suggestions and observations provided to improve our manuscript. We have checked point by point all reviewer’s comments and made the corresponding modifications to the main text. The English has now been checked by a -native English speaker.

Reviewer: The existence of HMM and LMM forms seems to be caused by the presence of either DNA or RNA. How the formation of CA and nucleic acids complexes is possible and why CA would bind (non specifically) nucleic acids 

Response: We believe that this rather artefactual interaction with nucleic acids is unlikely to take place in vivo because of the previously confirmed subcellular localization of the protein (Jensen, E.L et al 2019. ISME J. doi: 10.1038/s41396-019-0426-8), but this interaction could be linked to electrostatic interactions with other cell components, such as other proteins or lipids and these interactions could exist in vivo. We addressed this in the results and in the discussion (page 10, lines 297-305): “Because of the characteristic subcellular localisation of the ι-CA towards the periphery of the plastid of photosynthetic eukaryotes [6,15] it is unlikely that the HMM-nucleic acids form occurs in vivo, and so could be an unspecific artefactual association of multiple ι-CA monomers together with nucleic acids. However, the possibility that the ι-CA could interact with other cellular components (e.g., other proteins, lipids) cannot be discarded, in particular because both LMM and HMM are active and catalyse CO2  protonation [6]. The nucleic acid-bound HMM might mimic other forms induced by interaction with other negatively charged surfaces, such as galactolipids [27] that are abundant in plastid membranes.”

Reviewer: All methods finally lead to reconstruction of 3D structure of the tetrameric complex based on secondary structure prediction, disorder prediction and homology modelling. Despite the fact that interfaces between domains of individual CAs in the complex were modelled and are described, there is no adjustment of interfaces studied by Molecular Dynamics or other simulation approaches. Did authors perform or do they think about a computational analysis of the final composition?

Response: The overall aim of the modelling approach was to propose an arrangement of the domains in the tetrameric structure by using data available from similar and homologous structures to explain the observed experimental data and particularly to fit the tetrameric molecule in the SAXS envelope. It was not our objective to build an extensively refined model or to explain mechanistic details that are related to the overall assembly without additional structural data. We therefore performed energy minimization on the interfaces of the assembled dimeric building blocks of the domains but did not apply molecular dynamics of the whole model. We might consider to further refine the model in the future if additional structural data (e.g. Crosslinking Mass Spectrometry, XLMS) are available.

Reviewer: The most important question is if the activity of the CA depends on the presence of all 4 domains composing tetra-domain protein. Did authors addressed such question or do they have any experimental result clarifying roles of particular domain for the function?

This is a very interesting point indeed. As the reviewer mentioned, one particular feature of the ι-CA is the presence of domain repetitions, which might vary from one to several. In diatoms, we observed that these repetitions are often of 2, 3 and 4 domains. We addressed in a previous published work the question whether CA activity is present in those variants containing fewer domains. Using the same domain variants as those used in the present work, we confirmed that CA activity is well maintained (Jensen, E.L et al 2019. ISME J. doi: 10.1038/s41396-019-0426-8). However, we did not study the role of each domain in the overall CA activity of the 4 domain – homotetrameric – ι-CA in this work, and indeed we intend to address this particular issue in the future, as well as the role of each domain in the metal ion coordination, using site directed mutagenesis of the amino acid residues involved in the putative active/metal-binding site. 

We added the following sentence to the discussion in the main text (page 10, lines 341-344): “The CA activity of the four-domain tpι-CA, as well as the variants constructs ∆1-2-3 and ∆1-2, has been previously confirmed [6], however, whether all domains contribute to the overall CA activity or to metal binding in a particular tetrameric conformation is still unknown and must be further investigated”.         

Reviewer: It is clear that there is no homology between linkers connecting individual domains. On the other hand authors could try to find homologues of linkers over all know protein sequences to find if this connectivity element is unique or used in similar cases. Please comment it.

The presence of linkers in multidomain proteins have often functional and/or structural significance. In our manuscript, we show that the linkers found between the different domains in the tpι-CA have very low homology between them, despite their short extension, and display also different degree of disorder. In addition, we also show that in the four domain-containing tpι-CA the linker at the C-terminus (i.e. the one linking domains 3 and 4) has higher predicted disorder. This agrees with the predicted flexibility of the protruding arms of the tetrameric envelope determined by SAXS, which is not present in the sequences linking domains 1 and 2, and domain 2 and 3.

In order to clarify whether this is a particular feature of the tpι-CA, we compared the nature of the linkers present in other four domain-containing ι-CAs from other species. The linkers from ι-CAs from other diatom species, including Fistulifera solaris, Cyclotella cryptica and Thalassiosira oceanica, also have low homology and the linker between domains 3-4 (see figures in the attached pdf file) is predicted to be more flexible than the other linkers. In contrast, linkers from homologous ι-CA having less domain repetitions do not have high disorder prediction. Thus, this suggests that the flexible linker in the C-terminal extremity, and thus the protruding arms of the “drone-like” tetramer, are a particular feature of the four-domain-containing ι-CA.

We added the following paragraph to the main text to mention this (Discussion, page 11, lines 354-361):

This particular linker is also predicted to be disordered in the ι-CA sequences from other diatom species with four-domains, including Cyclotella cryptica, Fistulifera solaris and Thalassiosira oceanica. However, it is absent in the C-terminal linkers from homologous sequences having less domain repeats (data not shown). This suggests that the protruding and flexible arms observed in the SAXS envelope from the tetrameric tpι-CA is a particular feature of the four-domain ι-CA and, in agreement with our proposed models (Figure 5b), does not exist in other homologous sequences with fewer domains.
